# Isolation of Bioactive Metabolites from Soil Derived Fungus-*Aspergillus fumigatus*

**DOI:** 10.3390/microorganisms11030590

**Published:** 2023-02-26

**Authors:** Harman Gill, Ellen M. E. Sykes, Ayush Kumar, John L. Sorensen

**Affiliations:** 1Department of Chemistry, University of Manitoba, Winnipeg, MB R3T 2N2, Canada; 2Department of Microbiology, University of Manitoba, Winnipeg, MB R3T 2N2, Canada

**Keywords:** antibacterial, N-formyl-4-hydroxyphenyl-acetamide, atraric acid, non-lichen

## Abstract

Fungi produce numerous secondary metabolites with intriguing biological properties for the health, industrial, and agricultural sectors. Herein, we report the high-yield isolation of phenolic natural products, N-formyl-4-hydroxyphenyl-acetamide **1** (~117 mg/L) and atraric acid **2** (~18 mg/L), from the ethyl acetate extract of the soil-derived fungus, *Aspergillus fumigatus*. The structures of compounds **1** and **2** were elucidated through the detailed spectroscopic analysis of NMR and LCMS data. These compounds were assayed for their antimicrobial activities. It was observed that compounds **1** and **2** exhibited strong inhibition against a series of fungal strains but only weak antibacterial properties against multi-drug-resistant strains. More significantly, this is the first known instance of the isolation of atraric acid **2** from a non-lichen fungal strain. We suggest the optimization of this fungal strain may exhibit elevated production of compounds **1** and **2**, potentially rendering it a valuable source for the industrial-scale production of these natural antimicrobial compounds. Further investigation is necessary to establish the veracity of this hypothesis.

## 1. Introduction

Naturally occurring bioactive compounds produced by microorganisms and their structural analogs have been a continuous source of new lead molecules in the pharmaceutical industry. There are almost 22,500 bioactive secondary metabolites from almost all living organisms, including eukaryotes (fungi, plants, and animals) and prokaryotes (bacteria and cyanobacteria) [1,2]. Fungi are the second most diverse (3–5 million species) kingdom on earth; almost 9000 bioactive compounds, ranging from anti-microbial, -cancer, -allergic, and -oxidant, are produced by fungi [3]. The most prolific producers of natural products are ascomycetes fungi (*Aspergillus*, *Fusarium*, and *Penicillium*) [3].

*Aspergillus* is one of the largest and most intensively investigated taxons of fungi. There are more than 180 species in this genus, including *A. fumigatus*, *A. flavus*, *A. niger*, and *A. terreus* [4]. *Aspergillus fumigatus*, a ubiquitous fungus, is widely distributed almost everywhere, ranging from the air to soil and even the International Space Station [5,6]. It plays an essential ecological role as a decomposer, recycling carbon and nitrogen sources [7,8], and a biological role by producing a broad array of secondary metabolites [9]. In total, *A. fumigatus* has demonstrated the potential to produce at least 226 bioactive molecules [10], and of these, 36 chemical structures have been putatively linked to biosynthetic gene clusters (BGCs) in the *A. fumigatus* genome [11,12]. A single BGC has the potential to produce structurally and functionally diverse bioactive compounds [13]. For example, the production of ferroverdins and bagremycins, two classes of metabolites with distinct bioactivities, comes from the same BGC. It has been observed that the activation of this biosynthetic pathway is determined by the availability of iron [14]. This is an example of how BGCs can form “superclusters” that function to biosynthesize two or more similar molecules [15]. Thus, it provides an opportunity to explore the chemical space of *A. fumigatus*.

In our continuing investigation of bioactive secondary metabolites, a strain of *A. fumigatus* was obtained from a soil sample underneath a lichen thallus collected in Manitoba. The crude extract of this fungus displayed antimicrobial activity in preliminary bioassays. The bioassay-guided fractionation of the extract has resulted in the isolation and identification of two compounds, N-formyl-4-hydroxyphenyl-acetamide **1** and atraric acid **2,** with yields of ~117 mg/L and ~18 mg/L, respectively. The structures of these compounds (**1** and **2**) were elucidated using spectroscopic analysis. Additionally, an antimicrobial evaluation of these compounds revealed that these compounds exhibited significant antifungal activity against five selected fungal strains and modest antibacterial activity against multi-drug-resistant bacterial strains. To the best of our knowledge, this is the first report of these compounds from *A. fumigatus*. Previous reports of N-formyl-4-hydroxyphenyl-acetamide (compound **1**) have only been from two sources (marine-derived fungus [16] and *Penicillium chrysogenum* [17]), both with yields of ~1–3 mg/L. Herein, *A. fumigatus* showed significantly higher yields of ~120 mg/L. More significantly, this appears to be the first report of atraric acid **2** being isolated from a non-lichen fungal strain. In this manuscript, we describe the production, isolation, structural elucidation, and bioactive profiling of compounds **1** and **2**.

## 2. Materials and Methods

### 2.1. Sample Collection and Fungal Isolation

The soil sample was collected, along with a sample of lichen thallus, on the McGillivray Trail (latitude 49.8053, longitude −95.2374), Southeast Manitoba, in June 2020. The sample of soil was incubated on potato dextrose agar (PDA), for seven days, at 28 ± 2 °C. One (S4) of the fungal isolates displayed inhibition of the growth of the neighboring fungi on the same Petri plate. Therefore, it was selected for further investigation through further subculturing on PDA plates to provide a monospore culture of the fungus putatively labeled S4.

### 2.2. Identification of Fungus

Approximately 100 mg of the mycelia from the active S4 fungal strain was extracted using an isolation kit (E.Z.N.A.^®^ Fungal DNA Mini Kit, Omega Bio-Tek, Norcross, GA 30071, USA). Taxonomic identification was achieved through DNA amplification and sequencing of the internal transcribed spacer (ITS) region using ITS1F and ITS4R primers (Appendix A). Sequencing was performed at Cancer Care Manitoba Research Institute, the University of Manitoba. A homology search of the sequencing results (with primers removed) was performed using the BLASTn program against the NCBI GenBank database. Reference strain sequences with the highest identity and query coverage score were downloaded from the NCBI database. A phylogenetic tree was constructed using the neighbor-joining (NJ) method in MEGA 7.0 software with 1000 bootstrap replications [18]. This process revealed that the isolated fungus (S4) was a strain of *Aspergillus fumigatus.*

### 2.3. Fermentation and Extraction

The monospore culture of *Aspergillus fumigatus* was cultured on PDA, at 28 ± 2 °C, for 7 days. Four pieces of mycelial agar plugs (0.5 cm × 0.5 cm) were inoculated into two separate 2 L Erlenmeyer flasks, each containing 1 L of potato dextrose broth (PDB). The culture was shaken (200 rpm) at 28 ± 2 °C for seven days. The culture broth (2 L) was filtered to remove the mycelia, acidified with HCl (pH < 2), and extracted twice with an equivalent volume of ethyl acetate (EtOAc)**.** The combined organic layers were dried (Na_2_SO_4_), filtered, and evaporated under reduced pressure to obtain a crude extract, which was then used in subsequent steps.

### 2.4. Isolation and Identification of the Metabolites

The crude extract (650 mg) was fractionated by passing through a pre-packed normal phase spherical silica (10 g, 60 μm) flash chromatography column (Biotage^®^ Sfär Silica) using the Biotage Isolera Prime 3.3.0 apparatus. The flow rate was set at 10 mL/min, and UV/VIS detection was recorded at 264 and 245 nm. Pure compound **1** was eluted in a gradient mixture of 5:95% of Methanol (MeOH): Dichloromethane (DCM), and pure compound **2** was eluted in 10:90% of MeOH: DCM to provide ~234 mg and ~36 mg, respectively. The identity of the compounds was confirmed using nuclear magnetic resonance (NMR) and liquid chromatography-mass spectroscopy (LC-MS). The complete characterization data is given below.

### 2.5. Antimicrobial Activities

Four multi-drug-resistant, clinically isolated bacterial strains (MDRS) and five environmental fungal strains (Table 1) were chosen to evaluate the broad-spectrum antimicrobial activities of compounds **1** and **2**. The fungal strains designated as WIN(M) were obtained from the Winnipeg collection at the University of Manitoba.

#### 2.5.1. Preliminary Assay of Crude Extract

The antimicrobial activity of the crude extract was evaluated using the well diffusion method [19]. The crude extract, at 200 μg, was screened, in vitro, for its antimicrobial activity against bacterial and fungal strains. These assays were replicated three times under aseptic conditions. Bacterial plates were incubated at 37 °C for 24 h, while fungal plates were incubated at 28 ± 2 °C for 72 h. The zone of inhibition was determined by measuring the clear zone around each disc, and the average diameter was calculated in millimeters (mm). The experiment was performed in triplicates.

#### 2.5.2. Antimicrobial Susceptibility Testing for Isolated Compounds

Compounds **1** and **2** and their synergistic effect were tested using the disk diffusion method according to Clinical Laboratory Standard Institute (CLSI) guidelines [20]. In summary, a single bacterial colony was grown overnight in Luria broth (LB) and sub-cultured 1/100 in fresh LB. The strains were allowed to grow for 5 h, with shaking, at 37 °C and then diluted in 0.85% sterile saline to achieve 0.5 McFarland standard using a Densichek (Biomerieux, Canada). A sterile cotton swab was immersed into the bacterial suspension, and excess fluid was removed. The bacterial swab was further streaked in three directions in Mueller–Hinton agar (MHA) plates to obtain confluent bacterial growth. The media surface was allowed to dry, followed by the placement of the sterile paper disks and the addition of compound **1** (50 μg/mL, 100 µg/mL, 150 μg/mL, and 200 μg/mL), compound **2** (50 μg/mL, 100 μg/mL, 150 μg/mL, and 200 μg/mL), and compound **1+2** (25 μg/mL, 50 μg/mL, and 100 μg/mL each) on the disks. Ciprofloxacin was used as a positive control. The samples were dissolved in methanol; therefore, it was used as a negative control. Plates were incubated at 37 °C for 16–20 h, depending on the bacterial strain, and the zone of inhibition was recorded in millimeters (mm). Three independent biological replicates were performed.

#### 2.5.3. Minimum Inhibitory Concentration (MIC) of Isolated Compounds

Broth micro-dilution MIC determination was performed according to the CLSI guidelines. In brief, single colonies of bacterial cultures were grown overnight in LB and then sub-cultured 1/100 in fresh LB with further incubation at 37 °C with shaking for 5 h. Cultures were then diluted in 0.85% sterile saline solution to achieve 0.5 McFarland standard using a Densichek (Biomerieux, Saint-Laurent, QC, Canada). The diluted bacterial culture was further diluted, to 1:50 in MHB, for inoculation, to obtain approximately 5 × 10^5^ colony-forming units per milliliter (CFU/mL). The tested bacteria were 2-fold serially diluted in MHB and incubated at 37 °C for 16–20 h, depending on the bacterial strain. Tests on compounds 1, 2, and 1+2 (synergy effect) were performed from a minimum concentration of 16 μg/mL to 512 μg/mL. Ciprofloxacin and methanol were used as positive and negative controls respectively. The assay was performed with three biological replicates.

#### 2.5.4. Antifungal Assay of Isolated Compounds

Compounds **1** and **2** from *A. fumigatus* were tested using a disk diffusion method [21]. Briefly, paper disks (5 mm in diameter) with varying concentrations (100 μg/mL, 200 μg/mL, 300 μg/mL, and 500 μg/mL) of compounds on each disk were inoculated at four symmetrical points of the PDA plate. A fungal disk (5 mm in diameter) was placed in the center of the plate. Plates were incubated at 28 ± 2 °C until the control fungal mycelium covered the plate. Mycelia plugs, cultured on PDA without any added compounds, were utilized as controls to calculate mycelial inhibition. The assay was carried out with three replicates.

Antifungal activity was evaluated by measuring the inhibition zone using the following formula:(1)Growth inhibition percentage=[(A−A1)/A]×100

Here, A and A1 represented the growth diameters (mm) of the fungal growth in control plates and fungal growth toward the paper disk (at minimum inhibition concentration), respectively.

## 3. Results

### 3.1. Fungal Isolation and Preliminary Selection

A total of six fungal isolates with different morphologies were successfully isolated from underneath the soil sample of the lichen thallus. One out of six isolates was observed to be stunting the growth of neighboring fungi on the PDA plate (Figure 1b). Therefore, the crude extracts of all isolates were screened for antagonistic activities against bacterial (*Acinetobacter baumannii* RND efflux pump-deficient strain AB258, *Acinetobacter baumannii* wild type ATCC 17978, *Pseudomonas aeruginosa* RND efflux pump-deficient strain PAO750, MRSA ATCC43300), and fungal strains (*Trichoderma citrinoviride*, *Aspergillus oryzae*, *Aspergillus* spp., *Ganoderma lucidum*, and *Pleurotus ostreatus*). The crude extract, S4, (from the same fungus that inhibited the growth of surrounding fungi) exhibited antibacterial activity, especially against MRSA and *Acinetobacter baumannii* RND efflux pump-deficient strain AB258, with 19 mm and 16 mm zones of inhibition (Figure 1d,e), respectively. Similarly, the crude extract showed strong antifungal potential with a larger zone of inhibition (36 mm) against *Aspergillus* spp. (Figure 1f). Therefore, the S4 strain was selected and analyzed in further experiments.

### 3.2. Identification of Fungus

To identify the fungal strain S4, DNA barcoding was utilized. The ITS fragment of approximately 600 bp was amplified and sequenced. NCBI BLAST analysis showed that the query sequence shared a high degree of homology with *Aspergillus fumigatus* (accession number: MT529131.1), having 99% query coverage and 98.51% sequence similarity. A phylogenetic tree was constructed using the fungal strain S4 ITS region along with a few closely related species, such as *Aspergillus ruber*, *Aspergillus fischeri*, *Aspergillus flavus*, *Aspergillus niger,* and other reference sequences (Appendix A) obtained from the GenBank. *Penicillium* sp. was used as an out-group. Phylogenetic analysis further confirmed that query sequence S4 clustered monophyletically with a strain of *Aspergillus fumigatus* MT529131.1 (Figure 2) with 100 bootstrap support. Thus, based on these results (Figure 2), the isolated strain, S4, was identified as Aspergillus fumigatus.

### 3.3. Fermentation, Extraction, and Isolation of Compounds

*A. fumigatus* S4 was grown in 2 L of PDB, at 28 ± 2 °C, for 7 days. The ethyl acetate extraction from 2 L culture broth yielded 650 mg of crude extract. The chromatographic fractionation of the crude extract afforded two main compounds: **1** (~117 mg/L) and **2** (~18 mg/L).

### 3.4. Structure Identification

The structures of compounds 1 and 2 were identified using nuclear magnetic resonance (NMR) and liquid chromatography-mass spectroscopy (LC-MS). In addition, the data (Appendix A) were compared with the literature [16,22].

Compound 1 (N-formyl-4-hydroxyphenyl-acetamide): Colorless crystals; 1 H NMR (MeOD, 500 MHz) δ 9.06 (1 H, s, H-2′′), 7.11 (2 H, d, J = 8.1 Hz, H-2′, 6′), 6.76 (2 H, d, J = 8.1 Hz, H-3′, 5′), 3.57 (2 H, s, H-2), 13CNMR (MeOD, 125 MHz); δ 173.5 (C-1), 163.4 (C-2′′), 156.3 (C-4′), 130.0 (C-2′, 6′), 124.1 (C-1′), 115.0 (C-3′, 5′), 41.6 (C-2); HMBC correlations: H-2/C-1′,-2′/-6′, -2′,-1; H-2′/C-2, -3′, -4’; H-3′/C-1′, -2′, -5′, -4′. The molecular formula of compound 1 (Figure 3a) is C9H9NO3, a molecular ion of [M-H] 178.0581 (179.0582 calculated) was observed with the negative ion mode of LC-MS.

Compound **2** (atraric acid): Colorless oil; ^1^HNMR (MeOD, 500 MHz); 6.13(1 H, s, H-6), 3.78 (3 H, s, H-10), 2.27 (3 H, s, H-8), 1.96 (3 H, s, H-9), 11.92 (ArOH-3), ^13^CNMR (MeOD, 125 MHz); 172.5 (C-1), 162.5 (C3), 159.7 (C-5), 139.6 (C-7), 110.2 (C-6), 108.4 (C-4), 103.5 (C-2), 50.6 (C-10), 22.9 (C-8), 6.7 (C-9). LC-MS was identical to the literature [22].

### 3.5. Antimicrobial Activities

#### 3.5.1. Antibacterial Activities

Compound **1** and **2** and an equimolar mixture of **1** and **2** (synergistic effect), as well as the crude organic exact, were tested for their antibacterial activity against *Acinetobacter baumannii* RND efflux pump-deficient strain AB258, *Acinetobacter baumannii* wild-type ATCC17978, *Pseudomonas aeruginosa* RND efflux pump-deficient strain PAO750, and MRSA. The results showed that compounds **1** and **2** inhibited the RND efflux pump-deficient bacterial strains (*Acinetobacter baumannii* and *Pseudomonas aeruginosa*) with a concentration between 50 μg/mL and 200 μg/mL. The synergistic effect of compounds **1** and **2** is almost similar when these compounds were tested individually. The maximum zone of inhibition for compounds **1** and **2** (19.1 mm and 10.2 mm) (Figure 4) was reported against *Acinetobacter baumannii* AB258 at a 200 μg/mL concentration. Although a zone of inhibition (19 mm) was observed for the crude extract against MRSA, the isolated compounds did not exhibit any activity against the same bacteria. The positive control, ciprofloxacin (25 μg/mL), demonstrated a maximum zone of inhibition (26.9 mm) against *Pseudomonas aeruginosa*. No zone of inhibition was observed for the negative control (methanol), for any bacterial strains.

Further, the minimum inhibition concentration (MIC) values were evaluated. The MIC values of compounds **1**, **2,** and **1**+**2** (Synergistic effect) and the crude exact were >100 μg/mL. However, the positive control (ciprofloxacin) showed a MIC value of 10 μg/mL.

#### 3.5.2. Antifungal Activities

The evaluation of the inhibitory effect of compounds **1** and **2** on five selected fungal strains was performed by comparing the average diameters (mm) of mycelial growth of the plates with and without (control) compounds [22]. The control group consisted of mycelia grown without any added compounds. The results showed that, in comparison to the control, compound **2** displayed strong inhibition against all of the selected strains, with inhibition ratios ranging from 13% to 87% (Figure 5). In contrast, compound **1** exhibited mycelial inhibition only against *Aspergillus oryzae* (~14%), *Ganoderma lucidum* (~20%), and *Pleurotus ostreatus* (~80%).

Furthermore, the highest inhibitory effect was observed against *Pleurotus ostreatus* (Figure 6), with inhibition ratios of 79% and 88% for compounds **1** and **2**, respectively.

## 4. Discussion

*Aspergillus* is a diverse genus of fungi that are known as prolific producers of structurally diverse classes of compounds [23], such as alkaloids [24,25,26], anthraquinones [27,28], drimane sesquiterpene lactone [29], sterigmatocystin analogs [30], and many more. Several natural products of *Aspergillus* are well known for their pharmaceutical potentials, such as lovastatin, as a cholesterol-lowering drug [31]; butenolide derivatives, as anti-inflammatories [32]; 3′-(3-Methylbutyl)-butyrolactone II, as antifungals [33]; and aurasperone H, as an anticancer agent [28]. These compounds have significant potential medicinal applications, while *Aspergillus* species have also been reported to produce mycotoxins, such as aflatoxins, ochratoxin A, and fumonisins [34]. Despite this, scientists are actively investigating the potential benefits of compounds produced by these species.

In this study, a single strain of *Aspergillus fumigatus* was successfully sub-cultured from a soil sample that was fortuitously collected with a sample of lichen thallus. After applying the soil sample to a PDA plate, it was observed that one of the fungal strains present appeared to inhibit the growth of other fungal strains present on the plate. This suggested that this specific fungal strain was producing some natural antifungal products that warranted further investigation. This strain of fungus (*labeled S4)* was selected to screen potential antimicrobial metabolites. The subculturing of this fungus from the original soil plate, and the subsequent sequencing of the ITS region and construction of a phylogenetic tree, revealed that this fungus (S4) shared ~99% sequence similarity (100 bootstrap value) with a strain of *Aspergillus fumigatus.*

In order to investigate the bioactive metabolites produced by the *A. fumigatus* strain, a large-scale (2 L) fermentation culture was prepared in PDB. The crude EtOAc extracts displayed prominent bioactivity against MRSA and *Acinetobacter baumannii* AB258 (19 mm and 16 mm zones of inhibition, respectively). Similarly, the crude extract showed strong antifungal potential, with the largest zone of inhibition (36 mm) against a stock strain of an environmentally isolated, unknown *Aspergillus* spp.

Subsequent bioassay (antifungal)-guided fractionation of this crude EtOAc extract over silica gel afforded two pure molecules: compound **1** (~ 117 mg/L) and compound **2** (~18 mg/L). The structures of compounds **1** and **2** were successfully identified using NMR and LC-MS, and the results were compared with literature data to confirm their accuracy. The spectroscopic characterization of these two isolated metabolites revealed that compound **1** was N-formyl-4-hydroxyphenyl-acetamide and compound **2** was atraric acid.

The bioactivity profiles of compounds **1** and **2** and a mixture of both (**1**+**2**) were evaluated against a panel of four clinically isolated, multi-drug-resistant bacterial strains and five non-pathogenic fungal strains (Table 1). The maximum zones of inhibition for compounds 1 and **2** (19.1 mm and 10.2 mm) (Figure 3) were reported against *Acinetobacter baumannii* AB258. These findings showed that compounds **1** and **2** exhibited inhibition only against RND efflux pump-deficient bacterial strains but not ATCC17978, suggesting that efflux might be an important method for the removal of these molecules from bacterial cells. The synergistic effect of compounds **1** and **2** was similar to those found when the compounds were tested individually. A zone of inhibition (19 mm) was also observed for the crude EtOAc extract against MRSA; however, neither **1** nor **2** displayed any inhibitory activity against this same strain of bacteria. This suggests that there was an additional compound present in the crude extract that was not recovered in the subsequent purification process.

We also observed the antifungal activity of **1** and **2** when assayed against a series of fungal strains. It was observed that atraric acid **2** displayed excellent bioactivity in inhibiting fungal growth and that compound **1** (N-formyl-4-hydroxyphenyl-acetamide) exhibited moderate activity. The mycelial growth of the five fungal strains that we assayed was significantly inhibited by compound **2**, ranging from 37% to 87% inhibition of the fungal growth depending on which strain was assayed. The inhibitory effect of compounds **1** and **2** against *Pleurotus ostreatus* was the highest, with inhibition ratios of 79.38% and 87.62%, respectively. The antifungal activity of **2** may provide some rationale for the biosynthesis of this molecule by both lichen and non-lichen fungi, as competition for resources against other fungi in the environment would be a significant challenge.

There have been only two previous reports of N-formyl-4-hydroxyphenyl-acetamide **1** being isolated as a natural product. Compound **1** has been described as being produced by a strain of an unknown, marine-derived fungus and was assayed for radical scavenging [16], where it displayed strong activity. A second report of the isolation of **1** was from a strain of *Penicillium chrysogenum*, where it displayed moderate cytotoxicity against Du145, A-549, and HeLa cell lines [17]. To the best of our knowledge, these are only two reports of the isolation of compound **1**. In these previous reports, the yield of **1** was observed in a range of 1–3 mg/L; however, our strain of *A. fumigatus* was producing considerably higher yields of ~120 mg/L in unoptimized growing conditions. The reasons for the high level of production of **1** are not immediately apparent but may be linked to the unique ecological niche from which this strain of *A. fumigatus* was isolated. The production of 1 may be an adaptation mechanism to being in the soil underneath a lichen thallus that is producing an array of secondary metabolites.

Atraric acid **2** has been previously reported as a metabolite in lichen fungi [22], and it has been reported that **2** can exhibit anti-cancer activity against prostate cancer, act as an antagonist of androgen receptors [35], and possesses antioxidant and antimicrobial [36] activities against Gram (+) microorganisms, *Candida dubliniensis*, and *Candida krusei*. We observed an unoptimized production level of ~18 mg/L of atraric acid **2** from our strain of *A. fumigatus*. This report appears to be the first observation of atraric acid **2** being produced in a non-lichen fungus. The production of **2** by this strain of *A. fumigatus*, isolated from beneath a lichen thallus, may represent a horizontal gene-transfer event from lichen to non-lichen fungi. This type of gene transfer could result in the acquisition of new traits or abilities by the recipient fungi, such as increased tolerance to environmental stressors or the ability to produce novel secondary metabolites. However, there is limited research on the occurrence and effects of horizontal gene transfer between lichen and non-lichen fungi. Further studies are needed to fully understand the mechanisms and implications of this phenomenon. However, offering conclusive proof of that hypothesis is beyond the scope of this manuscript.

## 5. Conclusions

In summary, this study describes the isolation, full structure elucidation, and bioactivity profiling of compounds **1** and **2**. We observed a significant amount of production of compound **1** (~ 120 mg/L) from our strain of *A. fumigatus*. Both compounds **1** and **2** showed modest antibacterial activity but did show potentially promising antifungal activity. This is the first report, that we are aware of, on the production of these metabolites from *A. fumigatus* and the first report of **2** from a non-lichen fungus. It is possible that this specific strain of *Aspergillus fumigatus* has the potential for increased production of certain metabolites, which could make it useful for the industrial production of natural antimicrobials. However, more research would be needed to confirm this and determine the feasibility of large-scale production.

## Figures and Tables

**Figure 1 microorganisms-11-00590-f001:**
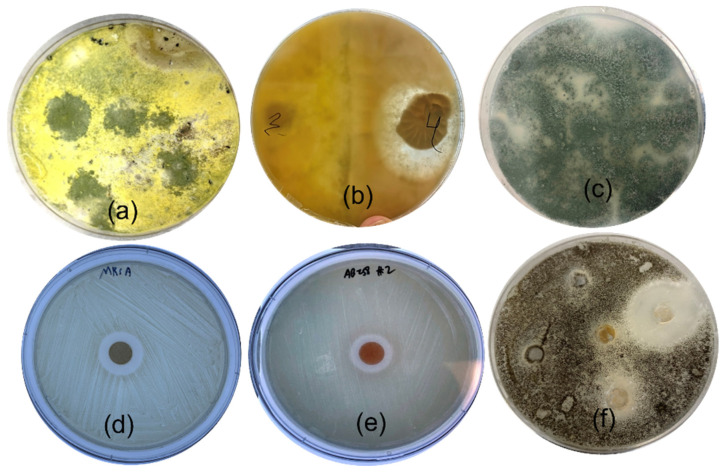
(**a**) Growth of fungi from soil samples on PDA plates; (**b**) inhibition of neighboring fungus by fungus S4; (**c**) monospore subculture of sample S4 fungus (*Aspergillus fumigatus*); (**d**) crude activity of the organic extract against MRSA; (**e**) crude activity of the organic extract against *Acinetobacter baumannii* AB258; (**f**) Antifungal activity of the crude extract against *Aspergillus* spp. Magnification is at 1× (*no magnification*).

**Figure 2 microorganisms-11-00590-f002:**
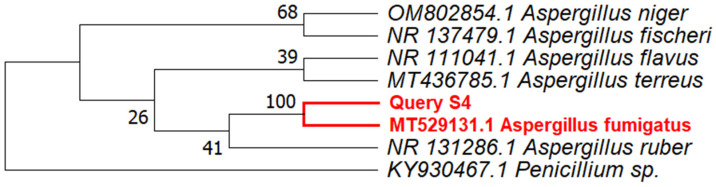
Phylogenetic tree showing the relationship of the ITS sequence of strain S4 with other fungal strains retrieved from the GenBank based on a BLASTn search. The tree was constructed using the neighbor-joining (NJ) method with MEGA 7.0 and using *Penicillium* sp. as an outgroup. Bootstrap values are based on 1000 replicates.

**Figure 3 microorganisms-11-00590-f003:**
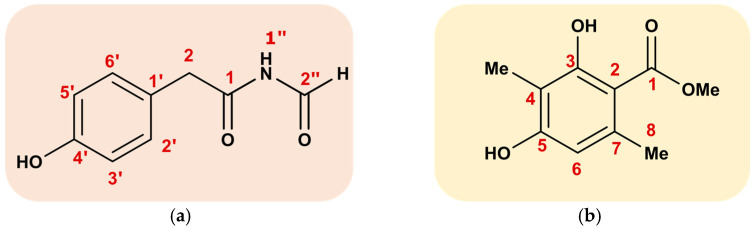
(**a**) Chemical structure of N-formyl-4-hydroxyphenyl-acetamide **1**, (**b**) chemical structure of atraric acid **2**.

**Figure 4 microorganisms-11-00590-f004:**
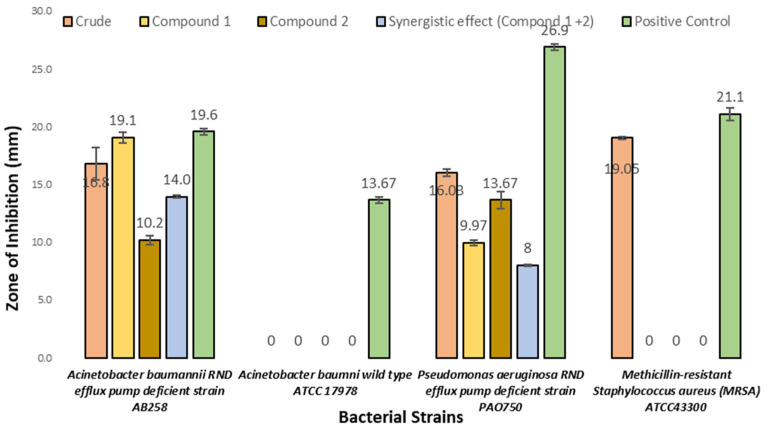
The average diameter of the zone of inhibition (mm) of the crude extract, compound **1**, compound **2**, the synergistic effect of compounds **1** and **2**, and the positive control (ciprofloxacin) against four clinical isolates of multi-drug-resistant bacterial strains. Data are presented as mean ± SD from three independent experiments, with a significance level of *p* < 0.05, versus the control group.

**Figure 5 microorganisms-11-00590-f005:**
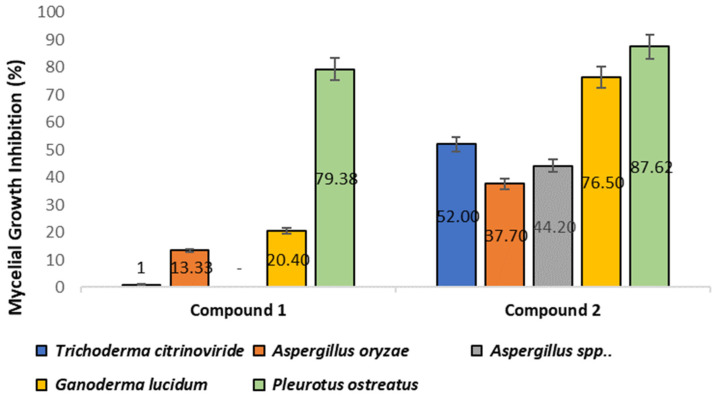
The percentage of mycelial growth inhibition of compounds **1** and **2** against five fungal strains, as compared to a control growth plate. The inhibition rates were calculated as mean ± SD (*n* = 3) and were considered significant at *p* < 0.05 in comparison to the control group.

**Figure 6 microorganisms-11-00590-f006:**
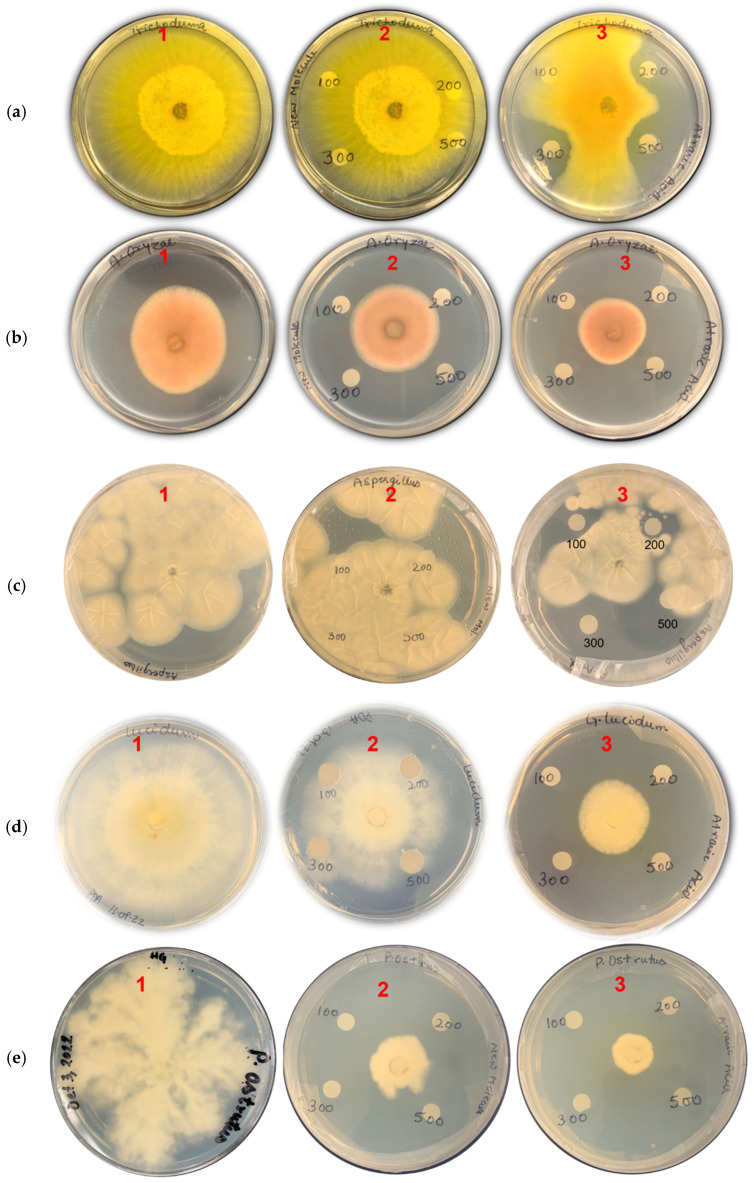
The antifungal activity of compounds **1** and **2** at different concentrations (100 μg/mL, 200 μg/mL, 300 μg/mL, and 500 μg/mL) on the mycelium growth of different fungal strains: (**a**) *Trichoderma citrinoviride*, (**b**) *Aspergillus oryzae*, (**c**) *Aspergillus* spp., (**d**) *Ganoderma lucidum*, (**e**) *Pleurotus ostreatus*; (1) control; (2) compound **1**; (3) compound **2**, Magnification is at 1× (*no magnification*).

**Table 1 microorganisms-11-00590-t001:** List of test organisms used for antimicrobial activities.

S. No	Bacterial Test Organism	Source
1	*Acinetobacter baumannii* (RND) efflux pump-deficient strain	AB258
2	*Acinetobacter baumannii* wild-type strain	ATCC17978
3	*Pseudomonas aeruginosa (RND) efflux pump-deficient strain*	PAO750
4	*Staphylococcus aureus* subsp. aureus Rosenach; methicillin-resistant	ATCC43300
	Fungal test organism	
1	*Trichoderma citrinoviride*	Lab strain *
2	*Aspergillus oryzae*	NSAR1
3	*Aspergillus* spp.	Lab strain *
4	*Ganoderma lucidum*	WIN(M)1804
	*Pleurotus ostreatus*	WIN(M)1803

* The University of Manitoba isolated lab strains confirmed via sequencing.

## Data Availability

Most of the data generated during this study are included in this article. The rest can be provided by the corresponding author upon request.

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
