# Peer review of "Isolation of Bioactive Metabolites from Soil Derived Fungus-Aspergillus fumigatus"

_microorganisms, 2023, doi:10.3390/microorganisms11030590_

Round 1
Reviewer 1 Report
Dears editor,
The present work is an interesting microbiology exercise. the manuscript should be improved in all sections of the manuscript.
In general, work is interesting and more-or-less well-structured. The principal problem is related to many discussion sections where authors will need to perform a major revision and update, and upgrade of the abstract, introduction, material and methods, results, and discussion.
Sincerely,
Author Response
Please see the attachmenet

Reviewer 2 Report
The article “Isolation of Bioactive Metabolites from soil derived fungus-Aspergillus fumigatus” by Gill et al., presented the isolation of two N-formyl-4-hydroxyphenyl-acetamide and atraric acid from soil derived fungus-Aspergillus fumigatus and tested their antifungal and antibacterial activities against various strains. This study is useful as these days scientists in different part of the world searching for active metabolites from microbes for different sectors. These metabolites can be used for several beneficial purposes. However, the article is missing significant information especially in material and methods such as information on statistical analysis and replication and repetition of the tests. Below points will improve the manuscript for its final publications.
It is important to mention the name of fungal strain in the title
Abstract is missing the statement regarding the significance of the study.
Mention the spectroscopic techniques used for the structural analysis of the isolated compounds in abstract part
Keywords must be different from those already used in title
Aspergillus fungus is also reported to produce carcinogenic compounds. This property should be brought in introduction or discussion part.
Is this the antimicrobial activity guided isolation of the active compounds?
A positive and a negative control are missing in antimicrobial activity tests
Information about Statistical analysis and repetition and replication of the experiment is not given
Source of fungal strains used in antifungal tests is missing
Discussion is shallow and somewhat diverge in several directions. Try to discuss the results with deep scientific depth and focus on your obtained results
As this is just the isolation of two active compounds and their activity test, this should be published as short communication rather than full article
Reviewer 3 Report
Gill and other authors presented a study where they identified an Aspergillus that produced secondary metabolites that are antimicrobial in nature. Authors have successfully isolated/identified two compounds that seem to have promise for future application.
With the sequence results of only 95% similarity achieved with your comparison to Aspergillus fumigatus, this seems a little low to state its one specific species with ITS. You should include results of other closely related fungi to A. fumigatus such as Aspergillus ruber and Aspergillus fischeri to strengthen your case for saying as such. Another problem maybe your sequencing quality as you have such “low” homology to the top hit. I would expect closer to 99% if it was same species. Also please include these more closely related Aspergilli in the tree as well as the bootstrap values on the tree in Figure 2. Might be worth considering sequencing another gene to be more confident in what species your Aspergillus is. As it stands, I am not wholly convinced this strain is A. fumigatus and not some other species of Aspergilli.
You should include a strain table for those used in your antimicrobial testing of compounds 1 and 2. The bacteria have a little more information since some have specific strain information, but this is completely lacking from fungal stains used. For instance, what is this Aspergillus spp that is used in the MIC testing throughout your experiments? Is this a lab strain or it a strain collected from the field?
With the focus on human associated pathogenic bacteria in the antimicrobial testing this is absent with the fungal testing. I would strongly recommend testing your compounds on some opportunistic fungal human pathogens like Candida albicans, Cryptococcus neoformans, or even other strains of Aspergillus fumigatus and include it in this paper if possible. The current fungi tested are not considered to be pathogenic in either plant of animals
Grammar notes for the manuscript, numerous times throughout the paper the spacing is off in the text. There are multiple instances of extra spaces and missing spaces between words in the text. Examples are extra spaces before sentences start or removal of space after “1” (example: compounds 1and 2). Both examples exist in the conclusion paragraph for reference.
Round 2
Reviewer 2 Report
Corrections have been done. Manuscript can be published